



# Flying UltraSonic - A new way to measure the wind

Martin Hofsäß[1], Dominique Bergmann[2], Jan Denzel[2], and Po Wen Cheng[1]

[1]Stuttgart Wind Energy @ Insitute of Aircraft Design (SWE), University of Stuttgart, Allmandring 5b, 70569 Stuttgart, Germany

[2]Insitute of Aircraft Design (IFB), University of Stuttgart, Pfaffenwaldring 31, 70569 Stuttgart, Germany

**Correspondence:** M. Hofsäß (hofsaess@ifb.uni-stuttgart.de)

**Abstract.** Measurements of flow conditions at complex sites that are difficult to install a met mast are expensive and can only be carried out with great effort. Concepts and new measuring methods are needed to evaluate these sites. This article presents an experiment in which an unmanned aerial vehicle (UAV), more precisely a helicopter, was equipped with a standard 3-D ultrasonic anemometer. This UAV was positioned closed to a meteorological measuring mast and remained stationary at a

constant altitude to measure the wind speed components. The data of the UAV were compared with the measurements of an ultrasonic sensor installed on the met mast. The measurements shows a deviation of $0.1\,\mathrm{m\,s^{-1}}$ for the horizontal speed. A comparison of the PSDs shows a very good agreement.

## 1   Introduction

The investigation of flow conditions in complex terrain is time-consuming and expensive. The installation of meteorological

measuring masts (met mast), which are equipped with in-situ sensors, is very cost-intensive and becomes more expensive with increasing the height of the masts. So-called lidar wind profilers have established themselves as an alternative to met masts. The lidar measurments show very good correlations with an slope of $0.9558$, an offset of $0.1577\,\mathrm{m\,s^{-1}}$ and a $R^2$ of $0.9984$ [Courtney et al. (2008)] in the flat terrain compared to in-situ sensors (e.g. cup anemometers). In complex terrain, however, there are considerable differences between the lidar measurements and the in-situ sensors of about $10\%$ [Hofsäß et al. (2018)].

Another disadvantage of ground-based lidar measurements is the poor reconstruction of turbulence and components of the Reynold stress tensor [Sathe et al. (2011, 2015)].

To capture detailed flow information in complex terrain, high-frequency and high resolution wind measurements are necessary. However, installations of a met mast is associated with high financial and planning costs. Alternative methods to measure the flow conditions are required. The use of unmanned aerial vehicles (e.g. aircraft and helicopters) represents a far more

favourable and flexible way of recording flow conditions. These unmanned aerial vehicles (UAV) can fly automatically (with the help of an autopilot system) or controlled by a pilot (a so-called remotely piloted aircraft system). If these UAVs fly automatically, the autopilot controls the previously defined waypoints and the autopilot flies the UAVs along this planned trajectory. The on-board computers of the test vehicles log the output of the installed sensors (e.g., the measured flow conditions) depending on the site coordinates, the position coordinates of the UAV and other flight parameters. The two different aircraft systems

helicopter and fixed wing aircraft differ in their flight capabilities. The fixed wing aircraft can, for example, fly a large area in





different flight patterns with high speed. The helicopter has its advantage in keeping a position during the flight. This results in different mission scenarios:

– The main strengths of the fixed wing aircraft lie in its long range and flight time, which enables the survey of large areas.

– Because of its hovering characteristics, helicopters are ideal for measuring fixed positions / heights.

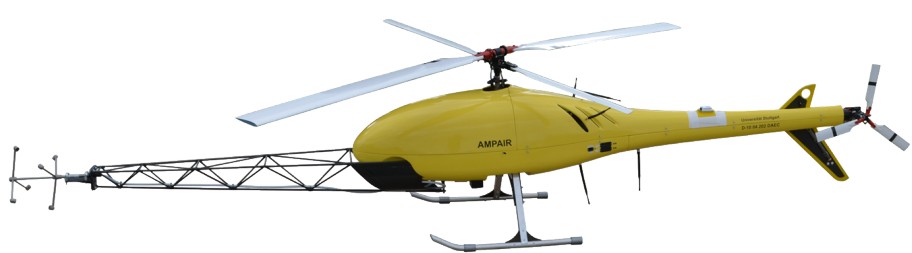

**Figure 1.** Flightsystem *AMPAIR* with installed boom and ultrasonic anemometer at the nose.

For this experiment an existing helicopter system called *AMPAIR* was used. This helicopter system *AMPAIR* was equipped with a 3-D ultrasonic sensor for the wind measurement. In comparison, a multi-hole probe is often used as a sensor for flow measurements in fixed wing aircraft [Wildmann et al. (2014, 2017)].

Such a combination of carrier system (helicopter) and measuring sensor (ultrasonic anemometer) has not yet been used for wind measurement. The aim of this experiment was to verify whether it is possible to determine the flow conditions with these alternative measuring methods. For the evaluation of the experiment data of a nearly located $95\,\mathrm{m}$ high meteorological mast (met mast) with in-situ sensors was used.

In section 2 the experiment is described with an introduction of the helicopter system *AMPAIR* and a description of the location. In the section 3 the data preparation is described and in 4 the results are presented. Finally, the conclusion in section 5.

## 2 Experiment description

The aim of this experiment was to investigate whether it is possible to measure the flow conditions with an unnamed helicopter. To evaluate the measurements, flow measurements from a nearby stationary wind met mast are used. The two different measurement systems are compared with each other using statistical values such as mean value and standard deviation of the measurements, and the data are also examined in the frequency domain. The following section is structured as follows:

In section 2.1 the used carrier system *AMPAIR* is introduced. Furthermore, in the section 2.2 the used measurement equipment is described, in the section 2.3 the site is characterized and in 2.4 the flight trajectory is presented.



## 2.1 AMPAIR Helicopter

*AMPAIR* is an electrically powered driven helicopter system, which was developed in the research project *SOGRO - Sofortret-tung bei Großunfall* [Bergmann (2013)]. The *AMPAIR* is shown in Figure 1 and the technical parameters of the UAV are shown in Table 1. Due to the large rotor diameter and the powerful electric drive, the helicopter has sufficient thrust to be able to carry
the sensors used for flow measurements, which are also used on wind met masts. The only question was how this sensor could be mounted outside the main influence area of the rotor downwash. Furthermore, the mounting structure should not influence the flight aerodynamics, and no vibrations from the helicopter should not affect the performance of the sensor. Various concepts were developed and the final design was realized using in light-weight beam construction. A sensor boom with a length of approx. $2.22\,\mathrm{m}$ with a weight of approx. $0.5\,\mathrm{kg}$ was developed and built. This was be manufactured by using carbon fiber.
This had the further advantage that the mounting structure is extremely stiff with very low weight. In this experiment a 3-D ultrasonic anemometer *uSonic-3 Scientific* from Metek was used. Due to vibrations and oscillations occurring during operation, the ultrasonic was not installed vertically as standard, but was mounted horizontally on the boom. Only the symmetrical arrangement of the ultrasonic anemometer on the longitudinal axis of the helicopter enables a nearly vibration-free operation of the sensor.

The flight time depends on the number of rechargeable battery packs carried. Since the maximum take-off mass is limited, a compromise must be found between the payload used (e.g., equipped sensors) and the flight duration (number of batteries). For this reason, the system is equipped with up to six battery packs and, depending on the flight time and required payload, it can be rearranged accordingly. One battery pack contains $11\,\mathrm{A\,h}$ each and has a mass of $4.8\,\mathrm{kg}$. This allows a flight time in between $10$ and $25\,\mathrm{min}$ with a variable payload of up to $10\,\mathrm{kg}$.

The *AMPAIR* is equipped with an autopilot, which allows waypoint navigation as well as fully automatic flights. In addition to an inertial sensor system, the helicopter also has an ultrasonic probe which is capable of measuring low altitudes for landing purposes. A telemetry connection to the system during the measurement flights allows a constant control of the flight and the adjustment of parameters in real time.

## 2.2 Measurement systems

This study compares data recorded with different measurement systems. To ensure that there is no time offset between the different data sources, both systems synchronized with GPS time. The system on the wind met mast is a central data acquisition system that collects the data centrally at a sampling rate of $50\,\mathrm{Hz}$.

Due to its size, the *AMPAIR* system does not need to be operated with miniaturized sensors, which, compared to commercially available sensor system (e.g., so called FirstClass sensors) show a larger deviation from the beginning. The used sensor
in this campaign was a 3-D ultrasonic anemometer *uSonic-3 Scientific* from Metek, which is also used on met masts. The use of proven and tested sensors that have been on the market for a long time should prevent negative influences from the measuring sensor. The *AMPAIR* contains two independently operating systems, to collect the required sensor data: The payload and the autopilot.





**Table 1.** Characteristics of the *AMPAIR* helicopter

| Attribute | Value |
|---|---|
| Maximum take-off weight (MTOW) | 46.15 kg |
| Rotor | 2.97 m |
| Rotation speed | 780 rpm to 820 rpm |
| Powertrain, electrical | 10 kW |
| Payload | up to 9 kg |
| Rechargeable battery | 58 V lithium-ion polymer battery |
| Control | Pilot-in-Command Computer-in-Command |
| Positioning System | DGPS |
| Time of flight (TOF) | max. 25 min |
| Data Acquisition system sampling | 10 Hz |
| Payload system sampling | 10 Hz |

– **Payload**: The payload contains the main components of the measuring system. The components are the electronics of the 3-D ultrasonic anemometer, the inertial measurement unit (IMU), the payload computer system electronic of the DGPS and the telemetry unit. The sensor head of the 3-D ultrasonic anemometer is connected to its evaluation electronics in the payload box and allows data recording up to 20 Hz. The AMPAIR is a mobile flying system with six degrees of freedom.

In order to transfer the measured wind speed data from the fixed system of the helicopter into the global reference system of the met mast, the attitude is determined with the inertial measurement unit as well as the position via the DGPS. By recording accelerations and rotation rates via the IMU, the helicopter system's own movements can be recorded. Thus, wind speeds measured by the moving sensor can be corrected.

– **Autopilot**: The autopilots system logs all data of the autopilot system required for flight controller. These can be syn-

chronized with the data of the payload via the GPS time. The following payload parameters are required for the post-processing of the measurement wind data: Position, altitude, acceleration, and heading of the helicopter

## 2.3 Site condition

The experiment was carried out near Grevesmühlen in the district of North-West Mecklenburg about 17 km south of the Baltic Sea. The site is shown in detail in Figure 2. At the time of the measurement, three wind turbines were installed at the site and

one 95 m high met mast. The direct environment is not built-up and corresponds to an agricultural area with fragmented trees and bushes. Further west, at about 1.3 km distance, there is a small settlement, and further south, at approx. 800 m distance, a densely forested area. The site corresponds to flat sites according to the IEC 61400-12 standard [International Electrotechnical Commission (2005)]. On the met mast there are several sensors at different heights for measuring wind speed, wind direction,





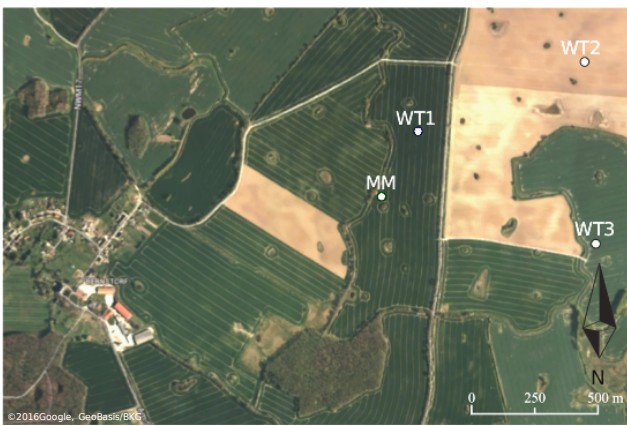

**Figure 2.** Overview of the test site with the three wind turbines (WT) and the met mast (MM).

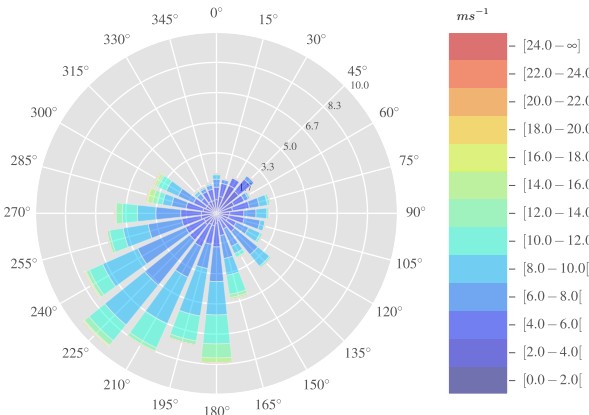

**Figure 3.** The Windrose of the test site.

pressure, and temperature (Table 2). The sensor used for the comparison was the *Thies Ultrassonic Anemometer 3D* at $93.2\,\mathrm{m}$. The wind met mast has a large undisturbed sector which is not affected by the three wind turbines. This free sector extends from $121.83°$ to $355.17°$ (based on standard IEC 61400-12). The main wind direction at this site is in the free stream sector and is in the south-west direction (see Figure 3). In addition to the information of the direction, the wind rose also shows the
5   wind speed bins in different colours. At this site wind speeds up to $14\,\mathrm{m\,s^{-1}}$ dominate. The data for the wind statistics were collected from 04.09.2013 to 18.05.2014.

## 2.4   Flight Plan

The flight plan for this experiment is shown in Figure 4. The flight starts near the position of the WEA1. This start was done by the pilot. The helicopter was flown near the previously defined position and altitude and then the control was transferred



**Table 2.** Summary of the installed sensors mounted on the met mast.

| Type of measurement | Height [m] | Sensor type |
|---|---|---|
| Wind speed | 95.5(hub height), 89.2, 59.2,40 | Cup anemometer |
| Wind speed & direction | 93.2 | 3-D Ultrasonic anemometer |
| Wind direction | 88.3, 39.0 | Wind vane |
| Rel. humidity | 92.0 | humidity sensor |
| Temperature | 92.0 | PT100 |
| Air pressure | 5.0 | Barometer |

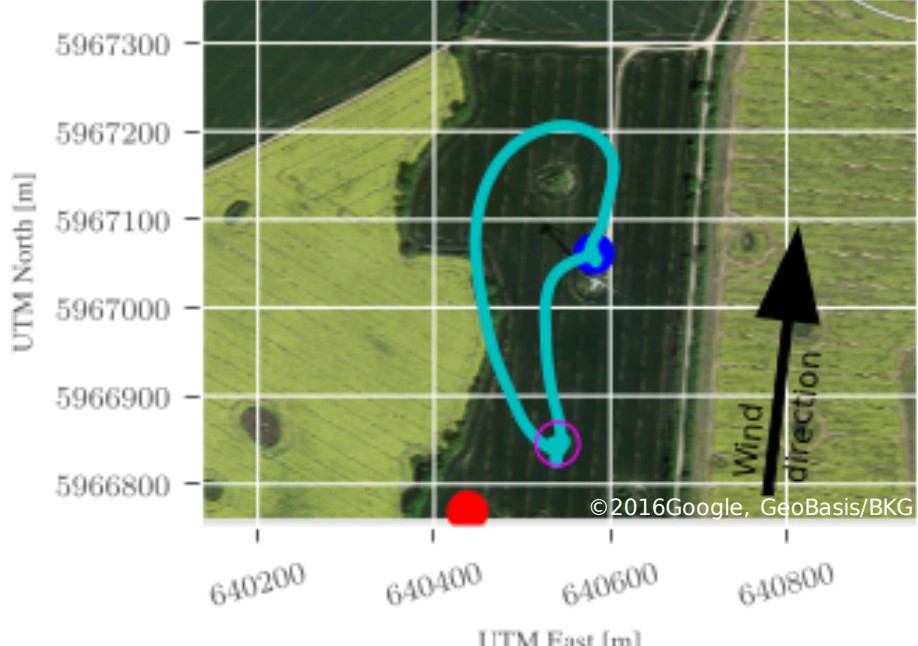

**Figure 4.** Overview of the trajectory with the start position (•), the measurement area (○) and the position of the met mast (•).

to the autopilot. The autopilot then flew the *AMPAIR* to the target position. After the mission time has expired, the pilot takes control again and flew the *AMPAIR* back to the landing site. Manual control was necessary because this feature has not yet been extensively tested on the autopilot and the risk of loss of control was too high. The average distance between the met mast and the helicopter position was $131.85\,\text{m}$ (Figure 5). Due to the influence of the wind, the autopilot had to adjust continuously

5  to keep the position. The standard deviation of the position signal was $1.696\,\text{m}$ and the range between the maximum and minimum distance is $\approx 8.5\,\text{m}$.

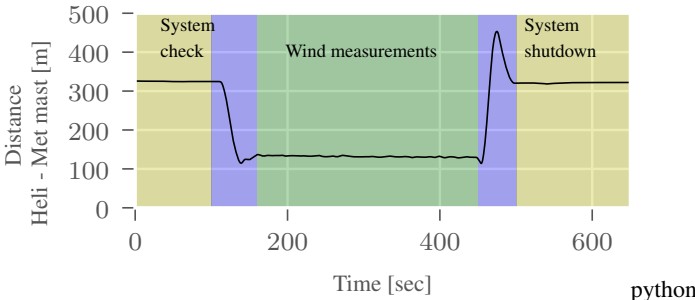

**Figure 5.** The different phases of the trajectory: In yellow is the phase of flight preparation and flight follow-up, in purple is the phase of approach and return and in green is the measurement phase.

## 3  Data analysis

This section describes the steps to compare the measured data of the helicopter with the reference data of the met mast. The necessary coordinate transformations are described in 3.1 and in section 3.2 the correction of the induced velocities is discussed. Induced velocities are additional velocities caused by the the helicopter's own motion, e.g. rotation around the body axes (see

section 3.2). Furthermore, the measured signal must be corrected by the airspeed.

### 3.1  Coordinate Transformation

In order to be able to compare the measured wind speeds from the met mast with the measurements from the helicopter, the data must be available in a common coordinate system. The coordinate system of the met mast (Global subscript $g$) is a right-hand coordinate system, the x-axis points to the north, the y-axis to the west and the z-axis to the sky.

The helicopter has two different coordinate systems: The one from the sensor (subscript $s$) and the one from the body (body subscript $b$). The two coordinate systems are right-hand systems. An overview of the individual coordinate systems is shown in Figure 6. In the sensor coordinate system the $Z_s$ axis points horizontally to the front, the $Y_s$ and $X_s$ axes are rotated by the mounting angle $\alpha$ to the horizontal plane. The $Z_b$ axes of the body coordinate system point downward, the $X_b$ points forward and the $Y_b$ points right. This is shown schematically in Figure 6b. The sensor coordinate system can be transferred to the body

coordinate system with the fixed mounting angle $\alpha = 30°$. Additionally the axes $X_s$ to $-Z_b$ and the axis $Z_a$ to $X_b$ are swapped, the wind speed components are also swapped accordingly. In the body coordinate system the $X_b$-axis (roll) shows along the rump, the $Z_b$-axis (yaw) is parallel to the shaft of the main rotor and shows downwards and the $Y_b$-axis (pitch) accordingly. The $ZY'X''$ convention is used to transform the components in the body coordinate system into the components of the global coordinate system. This means a rotation around the roll, pitch and yaw angles. The following steps are necessary:

– Rotation along the $X_b$ axis with the roll angle $\phi$ (b.1)

   – Rotation along the new $Y$ axis with the pitch angle $\theta$ (b.2)

   – Rotation along the new $Z$ axis with the yaw angle $\psi$ (b.3)





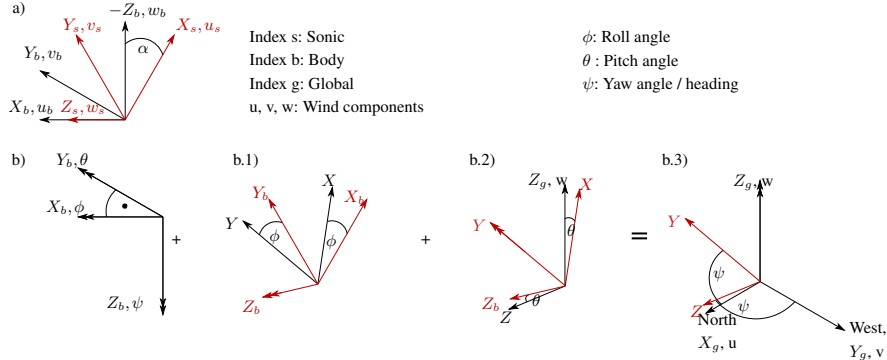

**Figure 6.** Schematic representation of the different coordinate systems. a: Dependency between sensor and body coordinate system. b: Transformation between the body and global coordinate system

$$
\mathbf{T} = \begin{pmatrix} \cos\phi & -\sin\phi & 0 \\ \sin\phi & \cos\phi & 0 \\ 0 & 0 & 1 \end{pmatrix} \cdot \begin{pmatrix} \cos\theta & 0 & \sin\theta \\ 0 & 1 & 0 \\ -\sin\theta & 0 & \cos\theta \end{pmatrix} \cdot
$$
$$
\begin{pmatrix} 1 & 0 & 0 \\ 0 & \cos\psi & -\sin\psi \\ 0 & \sin\psi & \cos\psi \end{pmatrix}
\tag{1}
$$

$$
\begin{pmatrix} u \\ v \\ w \end{pmatrix}_g = \mathbf{T} \cdot \begin{pmatrix} u \\ v \\ w \end{pmatrix}_b
\tag{2}
$$

This transformation is shown schematically in Figure 6 b.1 to b.3 and can be described mathematically by the equations (1)
5 and (2). From the components in the global coordinate system the wind direction (Equation (3)) and the horizontal wind speed (Equation (4)) can be calculated.

$$
dir = \arctan2(v, -u)
\tag{3}
$$

$$
v_{hor} = \sqrt{v^2 + u^2}
\tag{4}
$$



## 3.2 Induced velocities

Due to the fluctuating flow conditions, the autopilot must constantly readjust to maintain the predefined position. The self-motion induces additional velocities, which the ultrasonic also records. These induced velocities must be deducted from the recorded wind speed signal. The IMU of the helicopter records the rotation rates $\dot{\phi}$, $\dot{\theta}$, $\dot{\psi}$. These are needed to calculate the induced velocities. The starting point of the calculation is the transformation matrix $\mathbf{T}$ from equation 1. With the help of $\mathbf{T}$ multiplied by the direction vector from the center of the ultrasonic anemometer to the center of gravity of the *AMPAIR* $\boldsymbol{x} = [x,y,z]^T$ the movement of the ultrasonic anemometer in space can be described (Equation 5). The direction vector is $\boldsymbol{x} = [2.22, 0.0, -0.08]^T \mathrm{m}$. $\boldsymbol{x}$ is given in the rotating system. The induced velocities are the time derivative of the equation 5. The vector $\boldsymbol{x}$ is independent of time, the angles $\phi$, $\theta$, $\psi$ are time dependent. The representation of the individual terms of equation 6 is omitted here. The induced velocities calculated from Equation 6 are presented in the global coordinate system.

$$\mathbf{X} = \mathbf{T} \cdot \boldsymbol{x} \tag{5}$$

$$\dot{\mathbf{X}} = \frac{\partial (\mathbf{T} \cdot \boldsymbol{x})}{\partial t} = \left( \frac{\partial \mathbf{T}}{\partial \theta} \frac{\partial \theta}{\partial t} + \frac{\partial \mathbf{T}}{\partial \phi} \frac{\partial \phi}{\partial t} + \frac{\partial \mathbf{T}}{\partial \psi} \frac{\partial \psi}{\partial t} \right) \boldsymbol{x} \tag{6}$$

During the measurement phase, the rotation rates vary in the range of $\pm 0.1 \, \mathrm{rad\,s^{-1}}$ for the roll and yaw direction movement. The pitch rotation is above $\pm 0.2 \, \mathrm{rad\,s^{-1}}$ and thus twice as large as the other two rotations (Figure 7). In Figure 8 the calculated induced velocity components are shown in the global coordinate system. The induced velocities are in the order of $\pm 0.2 \, \mathrm{m\,s^{-1}}$ during the phase of wind measurement (stationary hovering flight). The induced speed has a very small influence ($\leq 0.1 \, \mathrm{m\,s^{-1}}$) on the $u$ component. The $v, w$ components fluctuate to a maximum of $\pm 0.2 \, \mathrm{m\,s^{-1}}$. Furthermore, the signal of the ultrasonic anemometer must be corrected by the airspeed. The flight speed of the helicopter is determined by the derived position signal (DGPS with a precision of $\pm 10 \, \mathrm{cm}$). All three components of the airspeed (Figure 9) are in the order of $\approx \pm 1 \, \mathrm{m\,s^{-1}}$ during the measurement phase. It would have been optimal if the autopilot of the *AMPAIR* had been able to keep its position fixed during the measurement phase. However, the instationary turbulent flow caused changing conditions, forcing the autopilot to readjust. Figure 9 also shows the flight manoeuvres ( purple area ), where the flight speeds are much higher than during hovering.

### 3.2.1 Effect of rotation on the sonic measurements

The ultrasonic anemometer was installed for this application in a horizontal position and not vertically as it is originally designed. In principle, the position should not make any difference when evaluating the wind speed components. However, there are considerable uncertainties as to which extent the ultrasonic anemometer will detect the induced rotational movements in the same way as assumed in this study. The functionality of a 3-D ultrasonic anemometer is simply described as follows: The ultrasonic anemometer has three transmitter-receiver pairs placed at $0°$, $120°$ and $240°$. Between the transmitter-receiver pairs there is a predefined distance. The transmitters emit sound waves and the receivers detect them again. The wind speed components $u$, $v$, $w$ are calculated from the different running times of the sound waves. The measuring principle should determine the

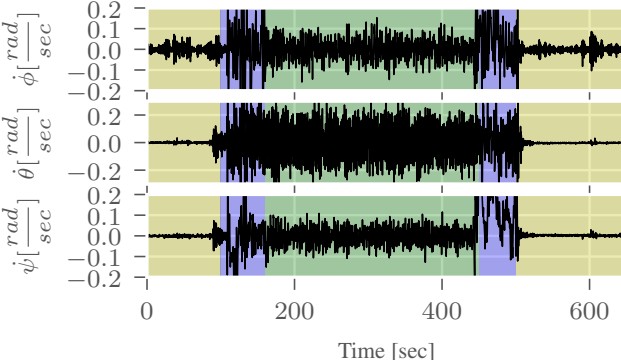

**Figure 7.** Rotation rate of the three directions (top graph Roll, middle Pitch, lower Yaw) in the body coordinate system.

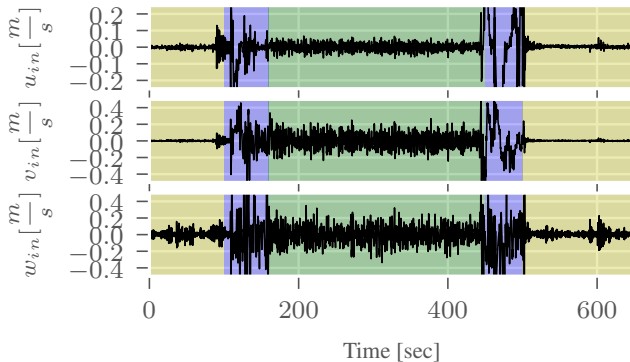

**Figure 8.** The three components of the induced velocities caused by the rotations due to the motion compensation of the auto-pilot in the global coordinate system.

correct velocity component for the translational movements. But the question is, what does the ultrasonic anemometer detect when it is rotated around its main axes? Unfortunately, no literature was found on this issue. However, considering the order of magnitude of the rotation rates and the resulting induced wind speeds, the possible error appears small during the hovering flight.

## 3.3 Workflow

This section 3.3 describes the workflow of data processing. In Figure 10 the schematic representation of the data processing is shown. The data of the *AMPAIR* are recorded with two different systems (AUTOPILOT, PAYLOAD). Both systems have a GPS-based clock, which is used to merge the data in the postprocessing. Both systems are sampling with 10 Hz. The merged data are then subjected to the coordinate transformation and corrected for the induced velocities. The data of the met mast are checked for plausibility. This means the sensors are compared with other sensors of the same type and checked for spikes. The sampling rate of the met mast data is 50 Hz. Finally the data is transformed into the common global coordinate system. The



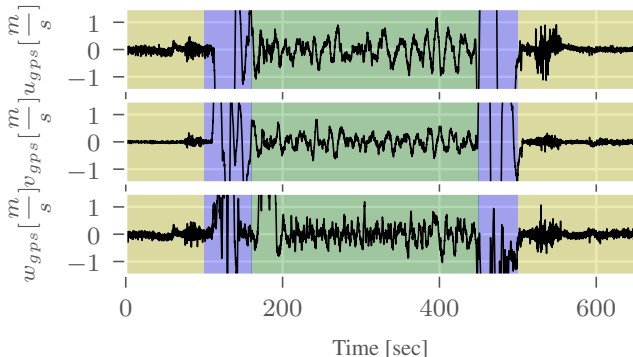

**Figure 9.** The three components of the induced velocities caused by translatory movement due to motion compensation of the auto-pilot in the global coordinate system.

recording of the data of the met mast was continuous and not only during the flight campaign. In order to be able to compare the data with each other, they must be synchronized a second time. For the evaluation only the time period is used at which the *AMPAIR* has reached its target position and has entered the stationary mode for measurement.

For some evaluations such as cross correlation or coherence it is necessary that both signals have the same time resolution, so the wind speed components are resampled from $10\,\text{Hz}$ to $50\,\text{Hz}$. A Fourier method is used [Hunter (2007)], the signal is assumed to be periodic. Alternatively, the higher resolution signal could have been sampled down to the lower frequency. In this case, information would have been lost in the higher resolution signal. It was investigated if it makes a difference, but there was no noticeable difference between up- and downsampling in the results of covariance and Pearson correlations coefficient.

To compare the two sensors, the mean values (marked as $\bar{x}$) and standard deviations ($\sigma$) are determined. Furthermore the turbulence intensity ($Ti$) (Equation 7), the turbulent kinetic energy (TKE) (Equation 8)

$$Ti = \frac{\sigma_{v_{hor}}}{\bar{v}_{hor}} \tag{7}$$

$$TKE = \frac{1}{2}\left(\sigma_u{}^2 + \sigma_v{}^2 + \sigma_w{}^2\right) \tag{8}$$

and the integral time scale ($T_{u*,v*,w*}$) (Equation 9) and integral length scale ($\Lambda_{u*,v*,w*}$) (Equation 10) are evaluated. Kundu [Kundu et al. (2002)] describe $T_{u*,v*,w*}$ as a scale of time over which $u*$, $v*$, $w*$ is highly correlated with itself and measure of memory of the process. The integral of the autocorrelation function $R_{u'u'}$ in equation 9 is evaluated until the first zero crossing (Kaimal and Finnigan (1994); Emeis (2018)). For the integral time and length scales the wind velocity components





are transformed in main flow direction (necessary condition $\bar{v}* = 0; \bar{w}* = 0$; values in the flow coordinate system are marked with $*$). The length scales in $v*$, $w*$ direction are evaluated with the wind speed $\bar{u}*$ in main flow direction.

$$T_u = \int\limits_0^\infty R_{u'u'}(\tau)\,d\tau \tag{9}$$

$$\Lambda_u = \bar{u}T_u = \bar{u}\int\limits_0^\infty R_{u'u'}(\tau)\,d\tau \tag{10}$$

The covariance (Equation 11 where $E[x]$ is the expected value of variable $X$) and the Pearson's correlation coefficient (Equation 12) provide additional values for the evaluation of the time series in relation to each other. The Pearson's correlation coefficient is in the range of $[-1, 1]$, where 1 means perfect linear correlation, 0 means no correlation and $-1$ means perfect negative linear correlation.

$$cov_{x,y} = E[x - E[x]] * E[y - E[y]] \tag{11}$$

$$\rho_{x,y} = \frac{cov_{x,y}}{\sigma_x\sigma_y} \tag{12}$$

To compare the data in the frequency domain, the Power Spectral Density ($PSD$) [Bendat and Piersol (2010); Hunter (2007)] for the wind speed components $u*, v*, w*$ is determined as well as the coherence ($C_{xy}$ Equation 13) between the components of the sensors.

$$C_{xy} = \frac{|P_{xy}|^2}{P_{xx}P_{yy}} \tag{13}$$

**4 Results**

This section compares the measurements between the met mast and the *AMPAIR*. The data are compared in the time and frequency domain. In the time domain the following quantities are statistically investigated: the mean and the standard deviation of three components of the wind speed, the resulting horizontal wind speed (Equation 4) and the wind direction (Equation 3). Additionally, the $Ti$ and the $TKE$ are compared. Also for the main flow components $u*$, $v*$, $w*$ the integrated time scale

$T_{u*,v*,w*}$ and the integral length scale $\Lambda_{u*,v*,w*}$ are estimated.

As a measure of how the time series correlate with each other, the covariance ($cov_{x,y}$) and the Pearson correlations coefficient $\rho_{x,y}$ are used. To compare the data in the frequency domain, the Power Spectral Density (PSD) [Bendat and Piersol





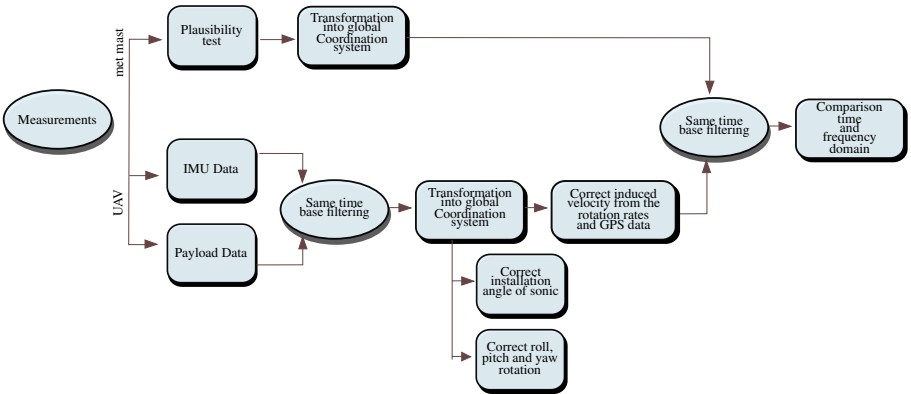

**Figure 10.** Overview of the data processing workflow, form the measuring of the different system to the comparising of the results.

(2010); Hunter (2007)] for the wind speed components $u*, v*, w*$ is determined as well as the coherence between the velocity components.

When evaluating the results, it should be noted that both measuring sensors are approx. $132\,\mathrm{m}$ apart from each other.

Before the results are looked at in detail, an overview is given of the distribution of the measurement data.

## 5 Distribution of the data

The distribution of the measurement data has a large influence on the statistical quantities such as $\sigma$, $Ti$ and $TKE$. The distribution depends significantly on how long the measurement is taken and whether the time series is stationary. The normalized probability density functions (PDF) of the $v_{hor}$ of the *AMPAIR*, that of the met mast ( evaluated time interval ) and for the whole $10\,\mathrm{min}$ period of the met mast are exemplarily presented in Figure 11. Furthermore the normalized normal distribution

is shown. The PDFs are normalized with $\sigma = 1$ and $v_{hor}^- = 0$. There are clear differences between the $10\,\mathrm{min}$ interval and the evaluated time interval. The distributions of the *AMPAIR* and the $10\,\mathrm{min}$ interval resemble to the normal distribution. Quite in contrast, the distribution of the met mast for the evaluation period does not. Especially in the range of $-3\sigma$ to $-2\sigma$ the distribution shows larger deviations compared to the normal distribution, also in the region of $0.5\sigma$. There is it more peaked. The evaluated time interval overestimate the statistical parameter like the $\sigma$. For the evaluated time interval $\sigma_{v_{hor}} = 1.61\,\mathrm{m\,s^{-1}}$

is compared to the $10\,\mathrm{min}$ period ($\sigma_{v_{hor}} = 1.442\,\mathrm{m\,s^{-1}}$) $11.6\,\%$ higher. The more unstable conditions make it more difficult for the *AMPAIR* to hover stationary and lead to more control interventions by the autopilot.

**Time domain**

Figure 12 shows the measured wind velocity components $u, v, w$ in the global coordinate system for the measurement phase. For a first comparison of the time series, the components of the three wind speed were visually analysed. These curves are not

perfectly aligned to each other, but show the same general trend. Looking at the $u$ component in detail, there are two significant drops in the wind speed at the signal from the met mast. This is not apparent in the data of the *AMPAIR*. The $v$ component



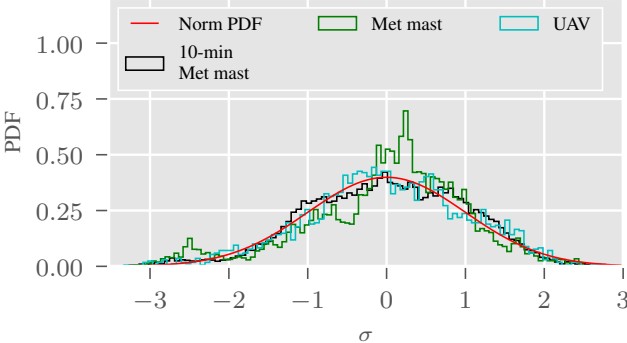

**Figure 11.** The Normed PDF for the $v_{hor}$ for a 10 min period and the period of the measurements.

shows a similar behaviour, the velocity of the signal from the met mast drops back several times to almost $0\,\mathrm{m\,s}^{-1}$. The signal from the *AMPAIR* fluctuates relatively constantly around the $-2\,\mathrm{m\,s}^{-1}$. The $w$ component of both sensors fluctuates around $0\,\mathrm{m\,s}^{-1}$. The fact that there was a distance of about approx. $132\,\mathrm{m}$ between the met mast and the UAV, the time series show a very good agreement. The investigated period represents the wind conditions at this site statistically very well. The mean wind

speed and the wind direction are in the region of the most typical wind conditions.

In addition to subjective observation, the data is also statistically evaluated. The mean value and the standard deviation are used as criteria here. The statistical parameters are listed in the Table 3 and Table 4. The mean value of the $u$ component confirms the previously gained impression. Both measuring systems measure approx. $8.3\,\mathrm{m\,s}^{-1}$, in percent the difference is only $1.12\,\%$ relative to the met mast data. The difference between the two signals for the $v$ component is $-1.147\,\mathrm{m\,s}^{-1}$ at

low wind speeds around $-0.966\,\mathrm{m\,s}^{-1}$. For the $w$ component, the difference is $-0.147\,\mathrm{m\,s}^{-1}$. If one considers the resulting horizontal velocity, the difference is $0.1\,\mathrm{m\,s}^{-1}$ and for the calculated wind direction ($v_{dir}$) the deviation between the two measuring systems is $7.817°$. The $u$-component show a very good agreement for the mean value, the $\sigma_u$ measured by met mast is $0.5\,\mathrm{m\,s}^{-1}$ higher than that measured by *AMPAIR*. $\sigma_v$ shows for the met mast a $0.18\,\mathrm{m\,s}^{-1}$ higher standard deviation compared to the *AMPAIR* and the $\sigma_w$ is nearly the same, which differ only by $0.03\,\mathrm{m\,s}^{-1}$.

Comparing the parameters for turbulence, such as $Ti$ and $TKE$, they show higher values for the met mast. $Ti$ is $6.5\,\%$ higher and the $TKE$ is $0.222\,\mathrm{m^2 s}^{-2}$ higher. By comparing the integral time scales $T_{u*}$ (Figure 13), the results differ significantly from each other. For the main wind direction is the $T_{u*} = 8.34\,\mathrm{s}$ for the met mast compared to $5.23\,\mathrm{s}$ for the *AMPAIR*. For $T_{v*}$ the value of the *AMPAIR* ($2.85\,\mathrm{s}$) is slightly higher than for the met mast ($2.50\,\mathrm{s}$). $T_{w*}$, however, is significantly larger for the *AMPAIR* ($4.38\,\mathrm{s}$) than for the met mast ($2.99\,\mathrm{s}$). The integral length scale $\Lambda_{u*,v*,w*}$ behaves in the same way as the time

scale. When comparing $\Lambda_{u*,v*,w*}$ with the values from the IEC standard 61400-1 for the design requirements of wind turbines [International Electrotechnical Commission (2014)] (Kaimal spectrum), the values show clear differences. The IEC standard indicates larger values for $\Lambda$ ($u* = 340.0\,\mathrm{m}$, $v* = 113.4\,\mathrm{m}$, $w* = 27.72\,\mathrm{m}$) compared to the estimated values from the met mast and the *AMPAIR*. The direct comparison of the time series show no linear correlation. The $\rho_{u*u*}$ value still has the largest match with $0.213$. The $w*$ component has no linear correlation ($\rho_{w*w*} = -0.008$) and the $v*$ component ($\rho_{v*v*} = -0.185$)





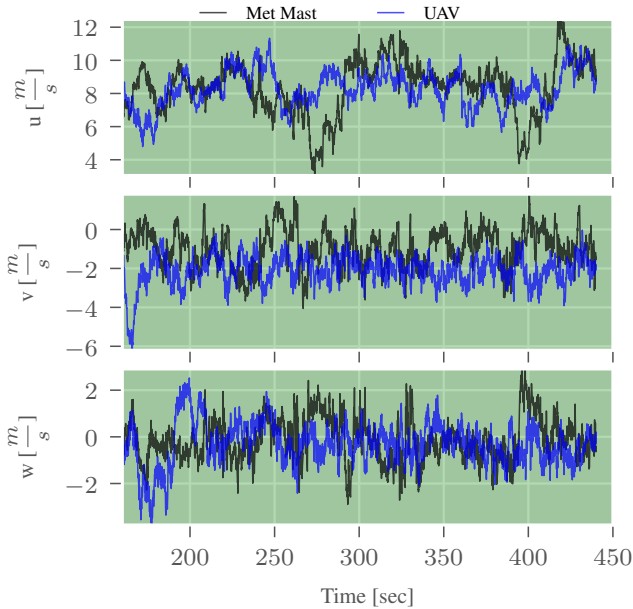

**Figure 12.** Time series of the wind components $u, v, w$ for both measurements systems during the measurement period. The met mast is in black and the *AMPAIR* signal in blue.

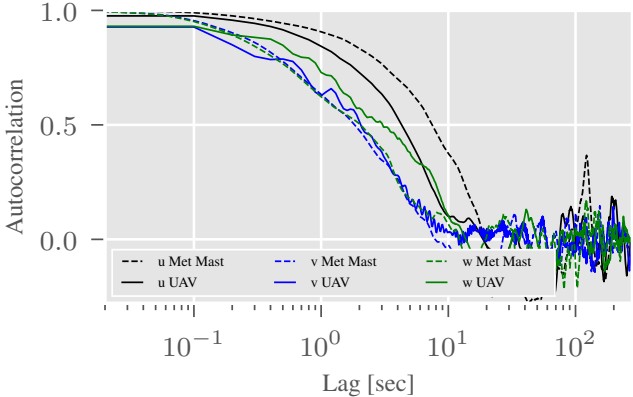

**Figure 13.** Autocorrelation of wind components $u, v, w$ for the met mast (dashed line) and the UAV (solid line).

shows slightly negative tendencies. The results give a conclusive picture. The integral length scale, as a quantity of how far the data correlates with itself, is much smaller as the distance between measurements systems. From this point of view, it is plausible that the values for $\rho$ show no correlation.





**Table 3.** The statistical values of the measurement phase for the UAV and met mast data

| Parameters | Met mast | AMPAIR | \| Deviation \| |
|---|---|---|---|
| $\bar{u}$ | $8.37\,\mathrm{m\,s^{-1}}$ | $8.28\,\mathrm{m\,s^{-1}}$ | $0.09\,\mathrm{m\,s^{-1}}$ |
| $\sigma_u$ | $1.61\,\mathrm{m\,s^{-1}}$ | $1.11\,\mathrm{m\,s^{-1}}$ | $0.50\,\mathrm{m\,s^{-1}}$ |
| $\bar{v}$ | $-0.97\,\mathrm{m\,s^{-1}}$ | $-2.11\,\mathrm{m\,s^{-1}}$ | $1.14\,\mathrm{m\,s^{-1}}$ |
| $\sigma_v$ | $0.96\,\mathrm{m\,s^{-1}}$ | $0.78\,\mathrm{m\,s^{-1}}$ | $0.18\,\mathrm{m\,s^{-1}}$ |
| $\bar{w}$ | $-0.15\,\mathrm{m\,s^{-1}}$ | $-0.30\,\mathrm{m\,s^{-1}}$ | $0.15\,\mathrm{m\,s^{-1}}$ |
| $\sigma_w$ | $0.90\,\mathrm{m\,s^{-1}}$ | $0.93\,\mathrm{m\,s^{-1}}$ | $0.03\,\mathrm{m\,s^{-1}}$ |
| $\bar{v}_{hor}$ | $8.48\,\mathrm{m\,s^{-1}}$ | $8.58\,\mathrm{m\,s^{-1}}$ | $0.10\,\mathrm{m\,s^{-1}}$ |
| $\sigma_{v_{hor}}$ | $1.61\,\mathrm{m\,s^{-1}}$ | $1.07\,\mathrm{m\,s^{-1}}$ | $0.54\,\mathrm{m\,s^{-1}}$ |
| $\bar{v}_{dir}$ | $186.65°$ | $194.46°$ | $7.81°$ |
| $Ti$ | $0.190$ | $0.125$ | $0.065$ |
| $TKE$ | $1.042\,\mathrm{m^2 s^{-2}}$ | $0.820\,\mathrm{m^2 s^{-2}}$ | $0.22\,\mathrm{m^2 s^{-2}}$ |
| $T_{u*}$ | $8.34\,\mathrm{s}$ | $5.23\,\mathrm{s}$ | $3.11\,\mathrm{s}$ |
| $T_{v*}$ | $2.50\,\mathrm{s}$ | $2.85\,\mathrm{s}$ | $0.35\,\mathrm{s}$ |
| $T_{w*}$ | $2.99\,\mathrm{s}$ | $4.38\,\mathrm{s}$ | $1.39\,\mathrm{s}$ |
| $\Lambda_{u*}$ | $70.26\,\mathrm{m}$ | $44.70\,\mathrm{m}$ | $25.56\,\mathrm{m}$ |
| $\Lambda_{v*}$ | $21.03\,\mathrm{m}$ | $24.35\,\mathrm{m}$ | $3.32\,\mathrm{m}$ |
| $\Lambda_{w*}$ | $25.18\,\mathrm{m}$ | $37.48\,\mathrm{m}$ | $12.3\,\mathrm{m}$ |

**Table 4.** The covariance and the Pearson coefficient for the UAV and met mast data

| Parameters | Quality |
|---|---|
| $cov_{u*,u*}$ | $0.368$ |
| $cov_{v*,v*}$ | $-0.146$ |
| $cov_{w*,w*}$ | $-0.008$ |
| $\rho_{u*,u*}$ | $0.213$ |
| $\rho_{v*,v*}$ | $-0.185$ |
| $\rho_{w*,w*}$ | $-0.010$ |

**Frequency domain**

To compare the data in the frequency domain, the Power Spectral Density (PSD) is calculated from the data. It should be mentioned again that the sampling frequencies of the two measuring systems differ (UAV: $10\,\mathrm{Hz}$; met mast: $50\,\mathrm{Hz}$). When calculating the PSD $S_{uu,vv,ww}$, efforts were made to take this into account and to adjust the resolution of the PSD. For the met mast, the PSD determined with $NFFT = 1024$ (NFFT: The number of data points used in each block for the FFT) data points and for the UAV with $NFFT = 256$. Furthermore, the time series were corrected by the respective mean value. In Figure 14




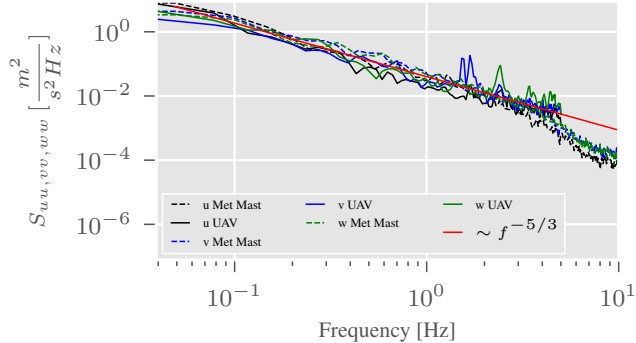

**Figure 14.** Power spectral density of wind components $u*, v*, w*$ for the met mast (dashed line) and the UAV (solid line).

the PSDs are shown in double-logarithmic scale. Additionally the theoretical spectrum for the ineratial subrange ($\sim f^{-5/3}$) [Pope (2000)]. In comparison, the PSDs of the individual components $u, v, w$ of the two measuring systems hardly differ from each other. The components $v, w$ of the UAV have clear spikes between $1.5$ to $3$Hz, which do not occur in the met mast data. The first assumption was, that these spikes were caused by the motion of the UAV (induced velocities), but these spikes are not

visible in the PSDs of the roll, pitch and yaw rotation rates and also not in the PSDs of the induced velocities. Also possible are oscillations, which are coupled in by the structure of the *AMPAIR*. However, it has not yet been possible to investigate this further. Definitely the influence of the rotor can be excluded because the rotor speed ranges from $780\,\mathrm{rpm}$ to $800\,\mathrm{rpm}$, which exceeds the measurable frequency range of the ultrasonic ($10\,\mathrm{Hz}$). The spectra of the *AMPAIR* show a qualitatively good course with the theoretical spectrum. The measurements of the met mast also agree well up to approx. $5\,\mathrm{Hz}$, at higher frequencies the

spectrum drops.

In order to investigate the dependencies of the time series of the wind speed components $u*, v*, w*$ in the frequency domain, the coherence (Figure 15) was calculated. In addition, the coherence model according to Pielke and Panofsky $coh(f) = exp(\frac{-a \cdot f \cdot D}{\bar{u}})$ [Pielke and Panofsky (1970)] was represented as a reference. The model uses as input parameters the frequency $f$, the mean wind speed $\bar{u}$, the distance of the measuring points $D$ and a decay parameter $a$ in the order of 10 . The values

here used are for the parameters $a = 1$ and $a = 10$ (see [Kristensen (1979)])and $D = 132\,\mathrm{m}$. For the parameter $a$ two different values were used, since the original proposed value ($a = 10$) appeared quite high compared to Simley [Simley and Pao (2015)]. Simley uses a modified model with a similar structure of terms, but only determined values between 0 to 2.5 for the parameter $a$ depending on $Ti$ and the stability conditions. The measured coherence shows no significant correlation between the sensors. The model after Pielke and Panofsky confirms this, that there should be no correlation at $132\,\mathrm{m}$ distance in the considered

frequency range.



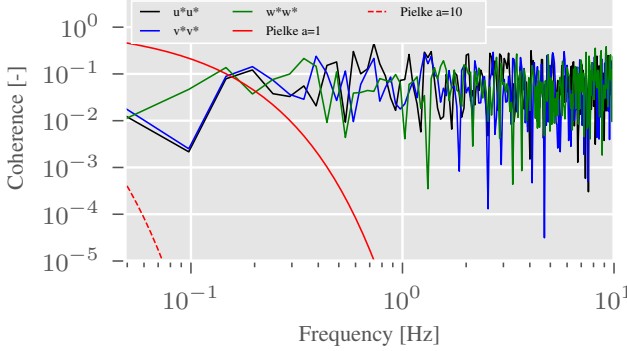

**Figure 15.** Coherence of wind components $u*$, $v*$, $w*$ for the met mast (dashed line),the UAV (solid line) and the reference model (red solid line).

## 5 Conclusions

In the previous sections, an experiment was presented in which an unmanned aerial vehicle equipped with a standard 3-D ultrasonic anemometer measured the flow components in stationary hovering flight. The comparison with a 3-D ultrasonic anemometer installed on a $95\,\mathrm{m}$ high met mast shows very good agreement of the horizontal wind speed with a deviation of

$0.1\,\mathrm{m\,s^{-1}}$ at mean wind speed of approx. $8.5\,\mathrm{m\,s^{-1}}$. For the wind component $u$ the differences are also very small. However, the $v,w$ components show larger deviations. For $v$ the deviation is $\approx 1.15\,\mathrm{m\,s^{-1}}$ and for $w \approx 0.15\,\mathrm{m\,s^{-1}}$.

When comparing the measured wind direction there are deviations of approx. $8.8°$. A possible cause is that the compass heading (Yaw alignment) is disturbed by the electromagnetic field of the main rotor motor. The compass heading is a very central quantity in the transformation of wind components from the helicopter coordinate system to the global coordinate

system. Another reason could be the wrong estimation of the $v,w$ components during the rotation of the UAV body.

The other statistical quantities (e.g. $\rho$, $cov$ ) show no similarities between the measurements. The reason for this is certainly the large spatial distance of $132\,\mathrm{m}$ and the relatively short common measurement period. The results of the integral length scales confirm this assumption. If the *AMPAIR* had come half closer to the met mast (distance of about $60\,\mathrm{m}$), the agreement should be better.

The comparison of the energy of the met mast and the UAV data in the frequency domain also shows a very good agreement among each other and also a very good agreement to the theoretical spectrum ($\sim f^{-5/3}$). Due to the low sampling of the UAV ultrasonic, no influence of the rotation of main rotor on the wind measurement can be determined.

The comparison of the coherence shows no dependencies between the wind speed components. However, the theoretical model does not show any more dependencies in the representable frequency range at a distance of $132\,\mathrm{m}$. For the representation

of the low frequencies, where there should be a correlation, a higher sampling rate and a longer flight time are necessary for a better resolution of the spectrum.

However, further improvements to the *AMPAIR* are needed. Sampling with $10\,\mathrm{Hz}$ is below the capability of the sensor ($20\,\mathrm{Hz}$). Furthermore, the effective measuring time must be increased. This is made possible by carrying more battery. However,





the additional weight has a negative effect on the flight duration. A doubling of the battery capacity does not mean a doubling of the flight time. Another possibility to increase the flight duration would be to convert the electric propulsion system to a combustion engine.

Despite the fact that the set-up is not yet perfect, this experiment shows that the principle of a flying ultrasonic anemometer
5 works. However, there is room for improvements. The primary goal of the optimization should be to extend the flight duration and to increase the sampling rate.

*Acknowledgements.* A part of this research was funded by the German Federal Ministry for Economic Affairs and Energy (BMWi) within the framework of the German joint research projects "Lidar complex" (Code number: 0325519). Special thanks to our pilot Jakob Straub for the successful measurement campaign and Tom Siebers for the great support.





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
