# Peer review of "Flying UltraSonic - A new way to measure the wind"

_Wind Energy Science, 2019_

## Referee Comment (RC1) · Anonymous Referee #1 · 13 Dec 2019

Review of wes-2019-81

Flying UltraSonic - A new way to measure the wind Martin Hofsäß, Dominique Bergmann, Jan Denzel, and Po Wen Cheng

Overview The manuscript entitled "Flying UltraSonic - A new way to measure the wind" by Martin Hofs{\"a}{\ss}, Dominique Bergmann, Jan Denzel, and Po Wen Cheng introduces a measurement platform combining a sonic anemometer point-measurement system fixed to a small helicopter UAV. The value of collecting reliable, high-frequency wind velocity vector and temperature data at locations typically inaccessible by other means cannot be overstated and would add tremendous capability to field measurement campaigns in atmospheric science, resource characterization for wind energy and complex flow studies. That said, the methods and data presented in the manuscript

suffer from serious faults that prevent the enclosed work from actually proposing a reliable method for making velocity measurements. Given that there does not appear to be any agreement between the fixed, met mast observations and the UAV measurements, the only conclusion available to the reader is that the system does not perform adequately, or that it was not measuring the same flow as the met mast. In either case, the manuscript as written does not provide any convincing arguments in support of the UAV-based measurement system.

Specific comments The introduction does not sufficiently frame the problem at hand. Airborne wind speed measurements are not a new innovation. Many systems have been integrated by the aerospace community for years. Nor are they new in the context of UAVs. The literature review in this respect requires improvement. Additionally the paper would benefit greatly from a more concise description of the novelty of the method, not to mention uncertainty, repeatability, etc.

The manuscript is not adequately organized. Tables and figures should be placed as close as possible to their first reference in the text. This would make the presentation of results in line with narrative and prevent readers from getting lost flipping pages.

UAV and met mast measurements cannot be said to agree comparing either the time series or statistical results. The authors state that this is due to the separation between the measurement locations, but the results do not inspire confidence in the results. The fact that spectra generally agree is not sufficient to say that the system makes accurate measurements of the atmospheric flow. Why is the separation between systems so large? How much improvement do the authors think would be available for the measurements with smaller separation?

How much uncertainty exists in the positioning system in terms of $\psi$, $\theta$, $\phi$ or $s$, $y$, and $z$? How would error in the positioning system impact the flow measurements?

Minor points Abstract The difficulty of measurements in complex sites is offered as a

main motivation for the development of the UAV-based method, but is not discussed at all in the paper. Either add some more discussion or consider removing references to complex terrain.

Measurement deviation of 0.1 m/s is mentioned, but without some sense of what atmospheric conditions this refers to, the deviation is not very helpful. Consider a relative measurement of error (i.e. XX%).

'... PSDs show very good agreement': this statement does not provide the reader with any sense of what is actually being compared.

Introduction Page 1 line 9: remove hyphen from 'time-consuming'

Page1 line 11: Remove 'So-called'. Profiling lidars are a mature technology and familiar to the atmospheric science and wind energy research communities.

Compound adjectives should be hyphenated throughout the text (e.g. high-resolution wind measurements, fixed-wing aircraft)

Page 2 line 5: is AMPAIR an acronym for something? Please define if so.

Section 2 Table 1: the maximum flight time is listed as 25 min. How does this depend on the atmospheric conditions? Given the stated 25 min flight time, why are test flights limited to 10 min? Is this due to power draw of the instrument/data acquisition system?

Figure 3: Consider adding a turbulence intensity rose in addition to the wind rose to more completely define the operating conditions at the test site.

Figure 4: poor image quality. Please replace.

Figure 5: why is there so much time for the system check and shutdown? Wouldn't more information be provided for a longer 'wind measurements' period? Why not extend to the 25 min limit of the system? Also, the figure says 'python' in the lower right corner. This doesn't seem related to the content of the figure.

Section 3 Page 7 line 21: why are Y and Z axis listed in red?

Equation (1): put full equation on a single line

Figures 7 & 8: what are the sources of noise in the system check and shutdown phases?

Page 10 line 6: Figure 10 appears 3 pages after in-text reference

Page 12: provide variable name in first text reference (e.g. line 5, 'The covariance, $cov$, . . .'). List for covariance, correlation coefficient.

Equation (13): $P_{xy}$, $P_{xx}$, $P_{yy}$ are not defined.

Section 4 Page 12 line 19: replace 'integrated' with 'integral'

Figure 11: caption should say 'normalized' rather than 'Normed'. What is the difference between 'met mast' and '10-min met mast' data? To which met mast data should the UAV data be compared?

Figure 12: time series do not appear to match well. Are there other samples for which agreement is better? How many samples of 8 min were collected? Some sense of the population statistics could support confidence in the measurement system.

Figure 13: correlation coefficients do not appear to match well except for $v$. Can the authors explain this figure more? How many time series were used to estimate $\rho$? Why do the results show such little agreement? Shouldn't $\rho$ for $v$ and $w$ approach 1 as the lag tends to 0 s?

Table 3: provide relative comparison of error rather than simply and absolute deviation. Otherwise it is not immediately clear how much this deviation matters.

Table 4: These results show that none of the measured velocity components are strongly correlated between the UAV and the met mast system. Don't these results indicate that either the UAV measurements are unreliable or that the two systems are

in fact measuring different flows? Either way, these results do not support the UAV system as a reliable measurement platform that can reproduce flow observations of the met mast.

Figure 15: Caption references a dashed line for the met mast, which does not seem to apply to the figure. The coherence between met mast and UAV signals appears to be bounded between $10^{-1}$ and $10^{-2}$ and do not agree with the theoretical formulation supplied. Again, these data seem to indicate that the measurements are unreliable or that the two systems are measuring different things.

Conclusions Page 18 line 6: deviation in $v$ is more than 1 m/s? This seems extremely large, even if the two systems are measuring different locations.

Page 18 line 11: The authors state ' other statistical quantities (e.g. $\rho$, $cov$) show no similarities between measurements.' What exactly is the purpose of this research if not to demonstrate agreement between the measurements? IF there is no agreement between the systems, how can readers have any confidence that the UAV system will be worthwhile?

Please also note the supplement to this comment:
https://www.wind-energ-sci-discuss.net/wes-2019-81/wes-2019-81-RC1-supplement.pdf

---

## Referee Comment (RC2) · Anonymous Referee #2 · 13 Dec 2019

The paper contains many grammar mistakes.

There is no discussion of calibration of the sonic anemometer with respect to induced flow from the rotor or from flow around the body of the aircraft in flight. The author states that the anemometer is mounted outside the "main influence" of the rotor wash, but it was not quantified how this was determined.

This type of measurement, especially using a rotary wing aircraft, is very power hungry. More discussion is needed to compare and contrast the benefits of long term measurments from a met tower with the spatial freedom and ease of deployment of a UAV.

Section 2.1, line 11 is unclear. perhaps provide a figure?

The paper discusses benefits to measurments in complex terrain, but the experiment is in simple terrain. Is this a starting point to more advanced measurements?

Pg 5 lin 8. Mentions WEA1 but then not referenced elsewhere.

Pg 5 lin 9 "Previously defined position" unclear

***The term airspeed is used repeatedly in terms of corrections. This is wrong. The 3D airspeed is measured by the sonic and the windspeed in a fixed coordinate system is the airspeed corrected by the groundspeed and aircraft rotations.

Pg 7 line 17: Rump?

Pg 7 - Use something other than red to denote a new frame

Pg 9 line 25 - does the sonic have the same measurment range and sensitivity in all 3 axes in order to justify turning it horizontally?

*** Rather than upsampling data assuming it's periodic, it's probably best to compare like signals at their lowest sample rate, 10 Hz.

Pg 13 line 20: Was convective time between measurement locations accounted for? Additionally, saying "same general trend" is a very ROUGH statement and debatable

***Pg 14 line 3 "Very good agreement" is debatable. There seem to be large deviations in secondary quantities like v, w and wind direction.

Pg 16 line 6: Provide justification, discuss affect on results.

Pg 17 line 2: "hardly differ" is a debatable point, given the log log format. Additionally, given statement on pg 16 line 6, did that cause the PSD to align better?

Pg 17 line 7: The arguement that the effect of rotor RPM can be excluded seems flawed. High RPM motion can still lead to lower frequency phenomenon.

Pg 17 line 9-10: What is the expected Kolmogorov scale and its relation to the dropoff in PSD?

Pg 19 line 4-5: Yes, the concept works. There seems to be much room for improvement.

Conclusions: Prior to focusing on duration and sample rate, focus should be placed on better correlation between measurments and additional investigation into induced flow influence due to the aircraft and its rotor.

---

## Author Comment (AC1) · 23 Jan 2020

**WES - Flying UltraSonic - A new way to measure the wind**
**Response to Reviewers**

M. Hofsäß et al.

January 23, 2020

**General Comments**

We thank the Editor and Reviewers for the rapid reviews and constructive comments.

Since receiving the reviews we have addressed the comments directly and modified the document. Please see the attached PDF for details of the changes.

**Reviewer 1**

The following is in response to Reviewer 1's comments. We thank Reviewer 1 for their rapid review and appreciate the recommendation.

**General comments**

*The manuscript entitled "Flying UltraSonic - A new way to measure the wind" by Martin Hofsäß, Dominique Bergmann, Jan Denzel, and Po Wen Cheng introduces a measurement platform combining a sonic anemometer point-measurement system fixed to a small helicopter UAV. The value of collecting reliable, high-frequency wind velocity vector and temperature data at locations typically inaccessible by other means cannot be overstated and would add tremendous capability to field measurement campaigns in atmospheric science, resource characterization for wind energy and complex flow studies. That said, the methods and data presented in the manuscript suffer from serious faults that prevent the enclosed work from actually proposing a reliable method for making velocity measurements. Given that there does not appear to be any agreement between the fixed, met mast observations and the UAV measurements, the only conclusion available to the reader is that the system does not perform adequately, or that it was not measuring the same flow as the met mast. In either case, the manuscript as written does not provide any convincing arguments in support of the UAV-based measurement system.*

Thank you for that assessment.

The total period of UAV measurements presented here is approximately seven minutes (e.g. Figure 12), and the UAV was approximately 130 m from the mast (page 13) in wind speeds around 8 m / s (15-20 seconds propagation time at these wind speeds). During this time the mean wind speed measured by the UAV and met tower were within 1.5% of each other (Table 3). Other mean statistics were within a few percent, while some of the higher order statistics showed higher deviation. Also, the PDF (Figure 11), autocorrelation (Figure 13), and frequency content of the two sets of measurements is very similar (Figure 14). This agreement suggests that the two systems were measuring the same flow and are capable of resolving the same frequency content.

The deviation of higher order statistics (e.g. in Table 3) and poor covariance (Table 4) suggests that the two systems were measuring the same flow but that the time series is simply not long enough to get better statistical agreement, given the distance and the length- and time scales prevalent in the atmosphere. This is supported by the low covariance (Table 4). We feel that this is an important demonstration of the potential capability of such a UAV-mounted system and a helpful contribution to the community, justifying this manuscript.

The poor correlations of the time series of the two measuring systems in the time and frequency domain (Pearson, Coherence) are no surprise. The studies of [Saranyasoontorn, K.; Manuel, L.; Veers, P. On Estimation of Coherence in Inflow Turbulence Based on Field Measurements. In Proceedings of the 42nd AIAA Aerospace Sciences Meeting and Exhibit; American Institute of Aeronautics and Astronautics: Reno, Nevada, 2004.] and [Simley, E.; Pao, L.Y. A longitudinal spatial coherence model for wind evolution based on large-eddy simulation. In Proceedings of the 2015 American Control Conference (ACC); IEEE: Chicago, IL, USA, 2015; pp. 3708–3714.] show similarly bad matches of the Coherence at spatial (lateral and longitudinal) offset of the measurement points. Saranyasoontorn used data from sonic anemometers which are installed on measurements masts with lateral offset and Simley used LES-Simulations. This work was not included in the original discussion. This has now been included in the discussion. Based on this work (Saranyasoontorn, Simley) and the statistical agreement between measuring mast and UAV, we hope that the reviewer agrees that – with some improvements – the UAV system would be beneficial for the research community.

**Major comments**

- *The introduction does not sufficiently frame the problem at hand. Airborne wind speed measurements are not a new innovation. Many systems have been integrated by the aerospace community for years. Nor are they new in the context of UAVs. The literature review in this respect requires improvement. Additionally the paper would benefit greatly from a more concise description of the novelty of the method, not to mention uncertainty, repeatability, etc.*

  We agree that the measurement of wind speed for aircraft is a solved problem, where e.g. Pitot-static tubes can be used to measure the speed of the aircraft. We also know that there are many aircraft-mounted devices can be used from aircraft to measure turbulence quantities for atmospheric science studies. However these typically need fixed-wing research aircraft, some of which are manned. Such measurements are also inherently expensive.

  Our motivation on starting this work in 2012 was to develop a system that could be used quickly an cheaply to carry out ad-hoc wind field measurements in the context of wind energy. This means in ambient wind speeds up to 25 or 30 m/s in the lowest 200 m of the atmosphere. We chose to focus on a helicopter system because it was available off the shelf, and could be landed without damaging the sensor. We note that at the time there were no comparable multi-copter designs that could have been used.

  Our goal in publishing this paper was to provide a record of the experiments that were carried out in 2013. We want to document our experiences so that others could benefit from them. It is our opinion that since then there have been no new comparable studies, and thus this is still a relevant contribution. Furthermore, we hope that the reviewer agrees that it is important to present and record results, regardless of whether the experiments were entirely successful or not.

  We have modified the paper to reflect this motivation.

- *The manuscript is not adequately organized. Tables and figures should be placed as close as possible to their first reference in the text. This would make the presentation of results in line with narrative and prevent readers from getting lost flipping pages.*

Thank you for this comment. We prepared the document using the WES LaTeX style and placed the figures and tables in the appropriate order. We chose not to modify the document to alter the positioning as this goes against the Journal's production guidelines. We presume that many of these layout challenges will be dealt with during the production process.

- *UAV and met mast measurements cannot be said to agree comparing either the time series or statistical results. The authors state that this is due to the separation between the measurement locations, but the results do not inspire confidence in the results. The fact that spectra generally agree is not sufficient to say that the system makes accurate measurements of the atmospheric flow. Why is the separation between systems so large? How much improvement do the authors think would be available for the measurements with smaller separation?*

Please see previous discussion.

The study of [Saranyasoontorn, K.; Manuel, L.; Veers, P. On Estimation of Coherence in Inflow Turbulence Based on Field Measurements. In Proceedings of the 42nd AIAA Aerospace Sciences Meeting and Exhibit; American Institute of Aeronautics and Astronautics: Reno, Nevada, 2004.] used stationary measuring masts and examined the lateral coherence. The results show very poor coherence for small lateral distances [7.7m, 15.8m, 38.2m]. The distance and the mean wind speed have a direct influence on the coherence: With higher wind speed ($> 15$m/s) and smaller distance the coherence increases, with lower wind speed ($<11$m/s) and larger distance the coherence decreases. For similar flow conditions as in our study, the coherence is less than 0.1 at the lowest frequency for a smaller lateral distance of only 38.2m.

In the study of [Simley, E.; Pao, L.Y. A longitudinal spatial coherence model for wind evolution based on large-eddy simulation. In Proceedings of the 2015 American Control Conference (ACC); IEEE: Chicago, IL, USA, 2015; pp. 3708–3714.], the longitudinal coherence is investigated using LES simulations at different distances and stability conditions. There it is shown that at 0.4 Hz the longitudinal coherence at the shortest distance studied (31.5m) is not coherence.

This facts were originally not considered in the discussion, but has now been added.

If the spatial offset could be reduced, we see clear potential here for improving results. To quantify the improvements with an absolute value would be untrustworthy, since this depends on many external factors (e.g. atmospheric stability, wind speed, measurement duration, measurement distance).

- *How much uncertainty exists in the positioning system in terms of $\psi$, $\theta$, $\phi$ or $s$, $y$, and $z$? How would error in the positioning system impact the flow measurements?*

This cannot be answered. The manufacturer of the autopilot does not provide any information on this. Especially for the conversion of wind speeds from the UAV coordinate system to the global coordinate system, the accuracy of the compass heading is of crucial importance. Errors in heading have a direct influence on the components $u$, $v$.

**Specific comments:**

- *Abstract: The difficulty of measurements in complex sites is offered as a main motivation for the development of the UAV-based method, but is not discussed at all in the paper. Either add some more discussion or consider removing references to complex terrain.*

  The motivation for the research project was to develop a measurement system which can also be used in complex terrain. The test flights in flat terrain were to validate the system. During the research project, we did not get permission from the responsible government agency to measure at the complex site.

- *Abstract: Measurement deviation of 0.1 m/s is mentioned, but without some sense of what atmospheric conditions this refers to, the deviation is not very helpful. Consider a relative measurement of error (i.e. XX).*

  Changed. This relative deviation has also been added to Table 3.

- *Abstract: '... PSDs show very good agreement' — this statement does not provide the reader with any sense of what is actually being compared.*

  The PSDs of the wind components match the theoretical spectrum in the inertial subrange very well. This has been updated.

- *Page 1 line 9 — remove hyphen from 'time-consuming'*

  Changed.

- *Page1 line 11 — Remove 'So-called'. Profiling lidars are a mature technology and familiar to the atmospheric science and wind energy research communities.*

  Changed.

- *Compound adjectives should be hyphenated throughout the text (e.g. high-resolution wind measurements, fixed-wing aircraft)*

  Changed.

- *Page 2 line 5 — is AMPAIR an acronym for something? Please define if so.*

  AMPAIR is a acronym for Autonomous Multipurpose Platform for Airborn Research. This is added.

- *Table 1 — the maximum flight time is listed as 25 min. How does this depend on the atmospheric conditions? Given the stated 25 min flight time, why are test flights limited to 10 min? Is this due to power draw of the instrument/data acquisition system?*

  The maximum flight time of 25min is only achieved if the complete payload is used for batteries. For safety reasons, these test flights were not flown with the maximum possible payload. Furthermore, the return flight was started with a safety reserve in the flight time. If the flight time had been fully exploited, it would have been possible to fly longer. Atmospheric conditions naturally also play a role. Air density plays a key role in performance. The denser the air, the less the engine has to perform and the more weight the helicopter can carry.

  However, it is energetically better for the helicopter if it makes a moderate forward flight and not only hovers. I'm not an expert in helicopter aerodynamic. For a closer look into this topic there a lot of books [Johnson, W. Helicopter theory; Dover Publications: New York, 1994; ISBN 978-0-486-68230-3.].

- *Figure 3 — Consider adding a turbulence intensity rose in addition to the wind rose to more completely define the operating conditions at the test site.*

  The turbulence intensity histogram has been added (new Figure 4).

- *Figure 4 — poor image quality. Please replace.*

  This Figure has been replaced with a new figure (see above). The flight path is now in Figure 2.

- *Figure 5 — why is there so much time for the system check and shutdown? Wouldn't more information be provided for a longer 'wind measurements' period? Why not extend to the 25 min limit of the system? Also, the figure says 'python' in the lower right corner. This doesn't seem related to the content of the figure.*

  Yes, we agree that longer measurements would have been better. However, because of the size and weight of the helicopter, careful attention has to be taken when starting. All systems (communication from ground station to helicopter, payload, autopilot, sensors, actuators, etc.) have to be checked. A similar procedure must be followed at the end of the measurement. These tests required battery use, and so precluded a longer flight time.

  We have deleted the word 'python'.

- *Page 7 line 21 — why are Y and Z axis listed in red?*

  The idea was to connect the text to Fig. 6 and to make the description easier to understand. Changed to black.

- *Equation (1) — put full equation on a single line*

  The form of the equation is adapted to the target format (2-column style). This can of course be undone, but would have to be adjusted again.

- *Figures 7 & 8 — what are the sources of noise in the system check and shutdown phases?*

  Figure 7: During the system check phase the sensors are tested by shaking the helicopter, and the wind also affects the helicopter. Shortly before take-off the rotors are activated (deflections just before the purple area). Please see comment to Figure 5.

  Figure 8: These are the induced wind speeds, which were calculated by the helicopter movement (Eqn. 6).

- *Page 10 line 6 — Figure 10 appears 3 pages after in-text reference*

  The positions of the figures was design for the 2 column style.

- *Page 12 — provide variable name in first text reference (e.g. line 5, 'The covariance, cov, ...'). List for covariance, correlation coefficient.*

  Changed.

- *Equation (13) — $P_{xy}$, $P_{xx}$, $P_{yy}$ are not defined.*

  Added.

- *Page 12 line 19 — replace 'integrated' with 'integral'*

  Changed.

- *Figure 11 — caption should say 'normalized' rather than 'Normed'. What is the difference between 'met mast' and '10-min met mast' data? To which met mast data should the UAV data be compared?*

Changed 'Normed" to normalized.

The 10 minutes refer to a typical period in wind energy over which the average is taken for power curves or load measurements. Met mast and UAV only refer to the time period when the UAV measured. It should be used to predict the used data, related to a longer averaging interval. Changed the description of the 10-minute dates and we hope that it is now more clearly described.

- *Figure 12 — time series do not appear to match well. Are there other samples for which agreement is better? How many samples of 8 min were collected? Some sense of the population statistics could support confidence in the measurement system.*

Following the comments of the reviewers, we have again made some changes in the analysis (see informations about the magnetic compass). The results show a better agreement in the mean values, but did not lead to improvements in the correlation studies (e.g. Pearson).

More measurement flights were made, but no comparable ones as described here. Sometimes the height was varied during the flight, sometimes the position. To measure to the 93m height, we slowly approached it and had no other flights on this height.

We think that additional flights would not make the results look better without reducing the distance UAV - met mast (compared to Simley et al. and Saranyasoontorn et al.).

- *Figure 13 — correlation coefficients do not appear to match well except for $v$. Can the authors explain this figure more? How many time series were used to estimate $\rho$? Why do the results show such little agreement? Shouldn't $\rho$ for $v$ and $w$ approach 1 as the lag tends to 0 s?*

From Table 4, the correlation coefficient for $v$ was -0.185 and with the new analyse the value is reduced to -0.077. For the estimation of $\rho$ only one time series is used and the time series is simply not long enough to get better statistical agreement.

This is not shown correctly in the graph. The problem is that the logarithm scale on the x-axis can not display the 0 lag. The smallest lag is 0.1 and this is the left limit in Figure 13. By using the linear scale for the x-axis, all lines started by the 0 lag at 1.

Changed the figure to linear scale in the document.

- *Table 3 — provide relative comparison of error rather than simply and absolute deviation. Otherwise it is not immediately clear how much this deviation matters.*

Add a extra column to Table 3 with the relative deviation.

- *Table 4 — These results show that none of the measured velocity components are strongly correlated between the UAV and the met mast system. Don't these results indicate that either the UAV measurements are unreliable or that the two systems are in fact measuring different flows? Either way, these results do not support the UAV system as a reliable measurement platform that can reproduce flow observations of the met mast.*

Statistically almost the same is measured, but in the high resolution time domain different events are measured. The change of the turbulent structures is too large over the spatial distance that Pearson and Covar can show dependencies at the two positions. This is supported by the results of Simley et al. and Saranyasoontorn et al.

- *Figure 15 — Caption references a dashed line for the met mast, which does not seem to apply to the figure. The coherence between met mast and UAV signals appears to be bounded between $10^{-1}$ and $10^{-2}$ and do not agree with the theoretical formulation supplied. Again, these data seem to indicate that the measurements are unreliable or that the two systems are measuring different things.*

  Changed caption. There was an error, the coherence is between the met mast and UAV. Please see Simley et al. and Saranyasoontorn et al., which use LES and met mast data and has also very bad coherence by much lower lateral distance as we had in our experiment.

- *Page 18 line 6 — deviation in v is more than 1 m/s? This seems extremely large, even if the two systems are measuring different locations.*

  The horizontal wind direction corresponds very well with a deviation of approx. 0.003 m/s but the wind direction and v-componente not, so the cause was intensively searched for. This was found in the signal of the magnetic compass. This shows an implausible behaviour during the system check phase. This fact was added and the evaluation and discussion was extended accordingly.

- *Page 18 line 11 — The authors state 'other statistical quantities (e.g. $\rho$, cov) show no similarities between measurements.' What exactly is the purpose of this research if not to demonstrate agreement between the measurements? If there is no agreement between the systems, how can readers have any confidence that the UAV system will be worthwhile?*

  See also previous discussion about the accuracy of the results.

  It is correct that $\rho$ and cov do not agree well. However, the mean values agree very well, so now the question is which characteristic value is more important for the evaluation of the results. If one compares the coherence or the covariance for 132m (longitudinal 89m; lateral 97m) distant measuring points and the consideration of Pielke, there is also no good agreement to be expected for the dynamic system wind. The Pielke model use only longitudinal distance.

  We changed the discussion to the results of Simley and Saranyasoontorn.

**Reviewer 2**

We thank Reviewer 2 for their rapid review and appreciate the positive recommendation.

The following is in response to Reviewer 2's comments.

**General comments**

*The paper contains many grammar mistakes.There is no discussion of calibration of the sonic anemometer with respect to induced flow from the rotor or from flow around the body of the aircraft in flight. The author states that the anemometer is mounted outside the "main influence" of the rotor wash, but it was not quantified how this was determined. This type of measurement, especially using a rotary wing aircraft, is very power hungry. More discussion is needed to compare and contrast the benefits of long term measurements from a met tower with the spatial freedom and ease of deployment of a UAV.*

*Section 2.1, line 11 is unclear. perhaps provide a figure*

In helicopter theory, a very similar model is used as the actuator-disk model in wind energy. The difference is that in the actuator disk model in wind energy, power is extracted and the helicopter model adds power. The power increases the flow speed after the rotor. Taking into account the continuity equation (same mass flow rate) the flow tube must be smaller than the rotor diameter. The boom extends approx. 1m further out than the rotor. In addition, the helicopter is slightly tilted forward, i.e. the downwash is directed backwards. The boom has a distance of about 1m to the tip of the rotor blade and think that this has very little influence on the measurements. To determine the influence of the rotor on the measurements is very costly. Tests on the ground do not work, because the stream cannot flow away freely (ground effect). This has to be tested in a wind tunnel, which is very difficult because of the size of the UAV. A significant influence of the rotor on the results cannot be observed. We now describe this issue in the paper.

A reference to Figure 1 was given, where the sonic is shown.

At the current state of the art, hovering UAVs are not yet suitable to replace stationary measurements with net masts over a long-term period. The primary problem will be the permanent power supply of the UAV. For short-term measurements ( $< 1\,\mathrm{h}$ ), hovering UAVs have great potential and can become an important tool in atmospheric research. Especially in rough terrain, e.g. over a wooded escarpment, they can become an important instrument for atmospheric research. However, only short periods of time are recorded, which can be sufficient for the validation of flow simulations or can provide important information for input variables of flow simulations. Furthermore, measurements at higher altitudes ($> 200\,\mathrm{m}$) could be of interest as a supplement to met masts at lower altitudes.

**Minor comments**

- *The paper discusses benefits to measurements in complex terrain, but the experiment is in simple terrain. Is this a starting point to more advanced measurements?*

  The motivation for the research project was to develop a measurement system which can also be used in complex terrain. The test flights in flat terrain were to validate the system. During the research project, we did not get permission from the responsible government agency to measure at the complex site.

- *Pg 5 lin 8. Mentions WEA1 but then not referenced elsewhere.*

  Changed. This is a mistake and should be labeled as WT1.

- *Pg 5 lin 9 "Previously defined position" unclear*

  Rewritten: The helicopter was flown close to the previously defined coordinates. We hope this make it clearer.

- *\*\*\*The term airspeed is used repeatedly in terms of corrections. This is wrong. The 3D airspeed is measured by the sonic and the windspeed in a fixed coordinate system is the airspeed corrected by the groundspeed and aircraft rotations.*

  Thank you for this point. We changed this and hope this is now much clear.

- *Pg 7 line 17: Rump?*

  This was a type error. Changed to main fuselage.

- *Pg 7 - Use something other than red to denote a new frame*

  Changed the color to black and bold and the arrows of the coordinate system are shown with a dashed line.

- *Pg 9 line 25 - does the sonic have the same measurement range and sensitivity in all 3 axes in order to justify turning it horizontally?*

  The Sonic anemometer has the same measuring resolution +-0.01 m/s in all three wind directions according to the manufacturer's specifications.

  Added in the paper: The manufacturer specifies the same measuring resolution for all three wind components.

- *\*\*\* Rather than upsampling data assuming it's periodic, it's probably best to compare like signals at their lowest sample rate, 10 Hz.*

  We agree with the author, both variants have been investigated. No difference could be found. We have done the evaluation again with a 10 Hz down sampled signal and rewritten the paper.

- *Pg 13 line 20: Was convective time between measurement locations accounted for?*

  No, the time lag between the different measurement position was not considered. The new evaluation with the down-sampled signal takes the time lag into account.

- *Additionally, saying "same general trend" is a very ROUGH statement and debatable*

  Changed and rewritten.

- *\*\*\*Pg 14 line 3 "Very good agreement" is debatable. There seem to be large deviations in secondary quantities like v, w and wind direction.*

  Yes, there are large deviations in the sizes like v and wind direction. Since the horizontal wind direction is very well matched with a deviation of approx. 0.003 m/s, an intensive search for the cause was carried out. This was found in the signal of the magnetic compass. This shows an implausible behaviour during the system check phase. This fact was added and the evaluation and discussion was extended accordingly.

- *Pg 16 line 6: Provide justification, discuss affect on results.*

  Changed. We use for the PSD the new down sampled signal and use the same number of points in the frequency domain.

- *Pg 17 line 2: "hardly differ" is a debatable point, given the log-log format. Additionally, given statement on pg 16 line 6, did that cause the PSD to align better?*

  Changed and rewritten.

- *Pg 17 line 7: The arguement that the effect of rotor RPM can be excluded seems flawed. High RPM motion can still lead to lower frequency phenomenon.*

  Changed and rewritten.

- *Pg 17 line 9-10: What is the expected Kolmogorov scale and its relation to the dropoff in PSD?*

  Rewritten, and using now the down samples signal for the evaluating

- *Pg 19 line 4-5: Yes, the concept works. There seems to be much room for improvement.*

  Yes, we agree. Please see also the next comment.

- *Conclusions: Prior to focusing on duration and sample rate, focus should be placed on better correlation between measurments and additional investigation into induced flow influence due to the aircraft and its rotor.*

  Rewritten, correction of the offset of the magnetic compass so that the statistical parameters (mean values) show a better agreement. The influence of the rotor on the measurements was also considered.

END